# Multiparameter Estimation with Two-Qubit Probes in Noisy Channels

**DOI:** 10.3390/e25081122

**Published:** 2023-07-26

**Authors:** Lorcán O. Conlon, Ping Koy Lam, Syed M. Assad

**Affiliations:** 1Centre for Quantum Computation and Communication Technology, Department of Quantum Science, Australian National University, Canberra, ACT 2601, Australia; 2Institute of Materials Research and Engineering, Agency for Science Technology and Research (A*STAR), 2 Fusionopolis Way, 08-03 Innovis, Singapore 138634, Singapore

**Keywords:** quantum metrology, collective measurements, entanglement, Cramér-Rao bound

## Abstract

This work compares the performance of single- and two-qubit probes for estimating several phase rotations simultaneously under the action of different noisy channels. We compute the quantum limits for this simultaneous estimation using collective and individual measurements by evaluating the Holevo and Nagaoka–Hayashi Cramér-Rao bounds, respectively. Several quantum noise channels are considered, namely the decohering channel, the amplitude damping channel, and the phase damping channel. For each channel, we find the optimal single- and two-qubit probes. Where possible we demonstrate an explicit measurement strategy that saturates the appropriate bound and we investigate how closely the Holevo bound can be approached through collective measurements on multiple copies of the same probe. We find that under the action of the considered channels, two-qubit probes show enhanced parameter estimation capabilities over single-qubit probes for almost all non-identity channels, i.e., the achievable precision with a single-qubit probe degrades faster with increasing exposure to the noisy environment than that of the two-qubit probe. However, in sufficiently noisy channels, we show that it is possible for single-qubit probes to outperform maximally entangled two-qubit probes. This work shows that, in order to reach the ultimate precision limits allowed by quantum mechanics, entanglement is required in both the state preparation and state measurement stages. It is hoped the tutorial-esque nature of this paper will make it easily accessible.

## 1. Introduction

Quantum parameter estimation involves preparing a quantum probe, allowing this probe to interact with the system we wish to learn about, and then examining the probe at the output. The maximum precision with which certain parameters can be estimated is dictated by the laws of quantum mechanics [1,2,3]. Quantum resources have been proposed as a way to improve measurement sensitivity in optical interferometry [4,5,6,7,8,9,10], quantum superresolution [11,12], quantum positioning [13,14], and tests of fundamental physics [15,16,17]. Several experiments have demonstrated enhanced precision estimation through the use of quantum resources [18,19,20,21,22,23,24,25,26,27,28,29,30,31,32,33,34].

Arguably, the full range of quantum mechanical effects is only revealed through multi-parameter metrology owing to the possible incompatibility of conjugate observables. There are many physically motivated reasons for studying multiparameter estimation. The simultaneous estimation of several parameters can enhance our ability to measure the different components of a magnetic field [35,36,37,38], multiple phase shifts [39,40,41,42,43,44,45], a phase shift and loss or phase diffusion [46,47], and can improve the tracking of chemical processes [48]. Additionally, quantum super-resolution for resolving two incoherent point sources of light [11,49,50] and many estimation problems concerning Gaussian quantum states [51,52,53,54,55,56,57,58] can be cast as multiparameter estimation problems. Given this motivation, there has been significant experimental [59,60,61,62,63] and theoretical [64,65,66,67,68,69,70,71,72,73,74,75,76,77,78,79,80,81,82,83,84,85] interest in quantum multiparameter estimation. See Refs. [86,87,88,89] for recent reviews on the subject.

The physical process of quantum metrology can be described as a quantum channel with several different variations possible depending on the degree to which we wish to exploit quantum mechanical effects [90]. In this work, we consider four different schemes; Classical-Classical (CC), Classical-Quantum (CQ), Classical with ancilla-Classical with ancilla ($C_{a}$$C_{a}$), and Classical with ancilla-Quantum with ancilla ($C_{a}$$Q_{a}$). These four schemes are distinguished by how the states are prepared, either using entangled states or non-entangled states, and by how they are measured at the output, as shown in Figure 1. Note that, on the state preparation side, we only consider either single-qubit or two-qubit states. This is distinct from previous studies that considered the case where entanglement is generated across all input states before the channel [91]. In our model, as the quantum states pass through the channel they experience a small rotation about all three axes by angles $\theta =(\theta_{x},\theta_{y},\theta_{z})$ before experiencing some form of decoherence. The quantum states are then measured, either using collective measurements or individual measurements. Individual measurements here mean that the probe states are measured one by one, in contrast to the most general measurement type, a collective measurement, which involves measuring multiple probe states simultaneously in an entangling basis. The quantum measurement strategies in Figure 1, thus refer to performing a collective measurement on asymptotically many copies of the probe state. However, we can also consider performing a collective measurement on a finite number of copies of the probe state. It is known that collective measurements offer no advantage over individual measurements (i.e., the CC and CQ strategies are equivalent, as are $C_{a}$$C_{a}$ and $C_{a}$$Q_{a}$) for estimating only a single parameter or for estimating multiple parameters with pure states [92]. However, for the most general metrology problem, multiparameter estimation with impure probe states, quantum resources can offer an advantage at both the state preparation and state measurement stages. The advantage of quantum resources at either of these stages can be thought of as quantum-enhanced metrology.

There are several known methods to determine fundamental limits on how accurately the parameters of interest can be measured. One common approach is to use the quantum Fisher information (QFI). There are several variants of the QFI, including the QFI based on the symmetric logarithmic derivative (SLD bound), introduced by Helstrom [93,94] and the QFI based on the right logarithmic derivative (RLD bound) [95]. It is known that for estimating a single parameter, the SLD bound can always be saturated, making it a particularly important bound. However, for estimating multiple parameters the SLD bound may not be attainable if the optimal measurements for estimating each parameter individually do not commute. An attainable bound on the ultimate achievable precision in quantum parameter estimation theory was formulated by Holevo [96,97], the Holevo Cramér-Rao bound. Throughout this paper, we shall simply refer to this as the Holevo bound. The Holevo bound is important as it is known to be asymptotically attainable when one uses a collective measurement on infinitely many copies of the probe state [98,99,100]. As such, the Holevo bound offers a way to investigate the precision attainable with the CQ and $C_{a}$$Q_{a}$ strategies in the asymptotic limit. In several different scenarios, using pure states and/or estimating a single parameter, the SLD bound and Holevo bound have been saturated experimentally [101,102] or there exist theoretical proposals to saturate these bounds [56,94]. However, it has recently been proven that if the SLD bound or Holevo bound cannot be saturated with individual measurements, then they cannot be saturated with any physical measurement [103]. This property is known as gap persistence and is an important caveat to the statement that the Holevo bound is asymptotically attainable.

In light of the practical difficulties associated with the SLD bound and Holevo bound, another bound of particular interest is the Nagaoka Cramér-Rao bound (Nagaoka bound) [104] which applies when one is restricted to measuring the probe states individually, i.e., the CC and $C_{a}$$C_{a}$ strategies in Figure 1. As it was originally introduced, the Nagaoka bound applies only to two-parameter estimation. This bound was later generalized to the *n*-parameter case by Hayashi [105,106], which we shall refer to as the Nagaoka–Hayashi bound (NHB) (For estimating one or two parameters, the NHB reduces to the Nagaoka bound and so in these scenarios, we shall use the two names interchangeably). The Nagaoka bound is known to be a tight bound for probes existing in a two-dimensional Hilbert space [107], i.e., there always exists a measurement that can achieve the same variance as the Nagaoka bound. However, it has since been shown that the Nagaoka bound is not always a tight bound [108]. Importantly, the Nagaoka bound or NHB will always give a variance smaller than or equal to that of the Holevo bound as individual measurements are a subset of collective measurements. Experimentally, the collective measurements required to surpass the NHB and approach the Holevo bound are extremely challenging to implement, hence there have been a very limited number of demonstrations of such measurements [109,110,111,112,113,114,115,116] (Also note that the techniques demonstrated in Ref. [117] could in principle be used to implement collective measurements). Recently, there has been a great deal of work developing new computational techniques for calculating the Holevo bound [56,118,119] and NHB [106].

This work aims to demonstrate the efficacy of these new techniques by finding the single- and two-qubit probes, which optimize the Holevo and Nagaoka bounds for several different noise channels. There have been several previous experimental and theoretical considerations as to how noisy channels affect the achievable precision [6,19,120,121]. In this work, we study specific examples of these noisy channels for qubit probes. We consider the problem of simultaneous estimation of three independent rotations around the *x*, *y*, and *z* axes of the Bloch sphere. We find a hierarchy between the four different schemes in Figure 1 in terms of the achievable precision subject to a noisy channel; $C_{a}$$Q_{a}$$\geq \{C_{a}C_{a},\mathrm{CQ}\}\geq \mathrm{CC}$, with no general ordering between $C_{a}$$C_{a}$ and CQ. Although one might expect that more entangled probes offer a quantum advantage over unentangled probes, this is not necessarily true. In very noisy channels probes with more entanglement can offer a disadvantage. As expected, it was also found that, for a noisy channel, collective measurements offer an advantage over individual measurements. Thus, we can consider these quantum mechanical effects as offering an increased robustness to noise. Typically, when estimating multiple parameters there is a trade-off between the number of parameters we wish to measure and the accuracy with which we can measure them [122,123]; however, we show for multiple-qubit probes this is not necessarily true. Some of the other, more surprising, features of quantum metrology are evident in the examples considered, for example, we find scenarios where states which experience decoherence outperform those which do not and we find discontinuities in the Holevo bound. We note that investigating these distinct schemes is different from many other works which have studied the scaling of the variance with the number of input probe states *N*, i.e., Heisenberg scaling (1/$N^{2}$) or scaling at the standard quantum limit (1/*N*) [26,48,124,125,126,127].

We begin by introducing the relevant bounds used in this paper in Section 2. In Section 3.1–Section 3.3, we discuss the results obtained for specific channels. For each channel, we consider single-qubit probes, two-qubit probes, and the attainability of the Holevo bound. In the appendices, we construct explicit measurement schemes which saturate the Nagaoka bound and NHB for many of the examples considered.

## 2. Preliminaries

### 2.1. Parameter Estimation and Quantum Fisher Information

The family of states being investigated, $\rho_{\theta }$, are parameterised by $\theta =(\theta_{1},\ldots ,\theta_{n})$, where the $\theta_{j}$ are the unknown parameters we wish to estimate. In this paper, the $\theta_{j}$ are qubit rotations on the Bloch sphere. We can measure the parameters we wish to estimate using a positive operator-valued measure (POVM). A POVM is described by a set of positive linear operators, $\Pi_{k}$, that sum up to the identity
(1)$$\sum_{k}\Pi_{k}=𝟙\;.$$

The *k*-th outcome is realized with probability $\mathrm{Tr}\{\rho_{\theta }\Pi_{k}\}$. Based on these measurement outcomes, we can construct an unbiased estimator for the parameters of interest, $\hat{\theta }$. Our estimated value is constructed from the estimator coefficients ξ
(2)$${\hat{\theta }}_{j}=\sum_{k}\xi_{j,k}p_{\theta }\left(k\right)\;,$$
where $p_{\theta }\left(k\right)$ is the probability of obtaining the measurement outcome denoted *k* and the sum is over all possible measurement outcomes. The aim of parameter estimation is to minimize the sum of the mean squared error between our unbiased estimate and the actual values we wish to measure, θ. For estimating several parameters simultaneously the mean-square error matrix, $V_{\theta }$, has elements given by
(3)$$\begin{array}{cc} {\left[V_{\theta }\right]}_{jk} & =\sum_{x}\left(\xi_{j,x}-\theta_{j}\right)\left(\xi_{k,x}-\theta_{k}\right)p_{\theta }\left(x\right)\;. \end{array}$$

If we have *N* independent and identical copies of the quantum state, $\rho_{\theta }$, the sum of the variance of our estimators is bounded by the quantum Cramér–Rao bound
(4)$$\mathrm{Tr}\left\{V_{\theta }\right\}\geq \frac{1}{N}\mathrm{Tr}\left\{J{\left(\rho_{\theta }\right)}^{-1}\right\}\;,$$
where $J(\rho_{\theta })$ is the QFI matrix with elements,
(5)$${\left[J(\rho_{\theta })\right]}_{jk}=\mathrm{Tr}\left\{\frac{\partial \rho_{\theta }}{\partial \theta_{j}}L_{k}\right\}\;,$$
and L is the quantum analogue for the classical logarithmic derivative. There is no unique way to define the quantum logarithmic derivative and each definition gives rise to a different QFI. Two of the most prominent QFI’s are those based on the symmetric logarithmic derivative (SLD) and the right logarithmic derivative (RLD). The SLD and RLD operators combined with Equations (Equation 4) and (Equation 5) give rise to the SLD bound, $C^{\mathrm{SLD}}$, and the RLD bound, $C^{\mathrm{RLD}}$, respectively. Although neither the SLD bound nor the RLD bound are attainable in general, both bounds are useful in certain scenarios. For this work, the SLD bound shall be used when we consider estimating a single parameter, as in this scenario it is known that the SLD bound is attainable. The SLD operators, L, can be computed as
(6)$$L_{k}=2\sum_{m,p}|e_{m}\rangle \frac{\langle e_{m}|\frac{\partial \rho_{\theta }}{\partial \theta_{k}}|e_{p}\rangle }{\lambda_{m}+\lambda_{p}}\langle e_{p}|\;,$$
where $|e_{m}\rangle ,\lambda_{m}$ are the eigenvectors and eigenvalues of the density matrix, $\rho =\sum_{m}\lambda_{m}|e_{m}\rangle \langle e_{m}|$, and the sum is over all $\lambda_{m}+\lambda_{p}\neq 0$ [128]. Thus, for any given problem the SLD QFI is generated in a completely deterministic manner meaning no optimization is required. As we shall only use the SLD bound for single parameter estimation, the corresponding bound on the variance in estimating the parameter $\theta_{k}$ is given by
(7)$$\begin{array}{c} v_{k}\geq C^{\mathrm{SLD}}=\frac{1}{J{\left(\rho_{\theta }\right)}_{kk}}\;, \end{array}$$
with *J* defined using the SLD operator, Equation (Equation 6).

### 2.2. Holevo and Nagaoka–Hayashi Bounds

Holevo unified the SLD bound and the RLD bound through the Holevo bound, which we denote H. The Holevo bound is achieved asymptotically and is assured to be at least as informative as $C^{\mathrm{SLD}}$ or $C^{\mathrm{RLD}}$, i.e., $H\geq C^{\mathrm{SLD}},C^{\mathrm{RLD}}$. The Holevo bound involves a minimisation over $X=(X_{1},X_{2},\ldots ,X_{n})$, where $X_{j}$ are Hermitian operators that satisfy the unbiased conditions
(8)$$\begin{array}{cc} \mathrm{Tr}\left\{\rho_{\theta }X_{j}\right\} & =0\;, \end{array}$$
(9)$$\begin{array}{cc} \mathrm{Tr}\left\{\frac{\partial \rho_{\theta }}{\partial \theta_{j}}X_{k}\right\} & =\delta_{jk}\;. \end{array}$$

The Holevo bound is
(10)$$\begin{array}{c} H:=\min_{X}\mathrm{Tr}\left\{Z_{\theta }\left[X\right]\right\}+\mathrm{TrAbs}\left\{ImZ_{\theta }\left[X\right]\right\}\;, \end{array}$$
where
(11)$$\begin{array}{c} Z_{\theta }{\left[X\right]}_{jk}:=\mathrm{Tr}\left\{\rho X_{j}X_{k}\right\}\;, \end{array}$$
and TrAbs{ImZθ[X]} is the sum of the absolute values of the eigenvalues of the matrix ImZθ[X]. *Z* takes the role of the inverse of the Fisher information matrix. The Holevo bound sets a limit to the sum of the variance of an unbiased estimate
(12)$$\begin{array}{c} \mathrm{Tr}\{V_{\theta }\}\geq H\;. \end{array}$$

Holevo derived this bound in Ref. [129], but the bound in the form shown above was introduced by Nagaoka [104]. A major obstacle preventing the more widespread use of the Holevo bound is that, unlike the RLD and SLD bounds which can be calculated directly, the Holevo bound involves a non-trivial optimization problem. However, as mentioned in the introduction, these difficulties have been somewhat alleviated in recent years [56,118,119].

For estimating two parameters, Nagaoka derived the bound
(13)$$\begin{array}{c} \mathrm{Tr}\left\{V_{\theta }\right\}\geq N:=\min_{X}\mathrm{Tr}\left\{Z_{\theta }\left[X\right]\right\}+\mathrm{TrAbs}\left\{\rho [X_{1},X_{2}]\right\}\;, \end{array}$$
valid if we are restricted to individual measurements. This bound is always more informative than or equal to the Holevo bound. As mentioned in the introduction, for two-dimensional systems, it is known to be attainable [104] and was conjectured by Nagaoka to be attainable in higher-dimensional systems as well [107]. For estimating more than two parameters with individual measurements we shall use the NHB [105,106]. The NHB can be computed from the following optimization problem [106]
(14)$$\begin{array}{c} \mathrm{Tr}\left\{V_{\theta }\right\}\geq N:=\min_{L,\;X}\left\{Tr\left[S_{\theta }L\right]\;|\;L_{jk}=L_{kj}\;\mathrm{Hermitian},\;L\geq XX^{⊺}\right\}\;, \end{array}$$
where $S_{\theta }=1_{n}⊗\rho_{\theta }$, $1_{n}$ is the n×n identity matrix (Note that a different notation is used for identity matrices which exist in the classical vector space, compared to those in the quantum Hilbert space as in Equation (Equation 1)) and $X={\left(X_{1},X_{2},\ldots ,X_{n}\right)}^{⊺}$ is a vector of Hermitian estimator observables $X_{j}$ that satisfy the locally unbiased conditions, Equations (Equation 8) and (Equation 9). The matrix $S_{\theta }$, exists on an extended quantum-classical Hilbert space. We use the symbol Tr[·] to denote trace over both classical and quantum systems. We shall use the symbol N to denote both the Nagaoka bound and NHB, and it will be obvious from the number of parameters being estimated which we are referring to. We can define the most informative bound as the precision achievable through individual measurements, $C^{\mathrm{MI}}$, such that for two-dimensional systems estimating two parameters $C^{\mathrm{MI}}=N$ [104]. As mentioned in the introduction, the NHB is not always a tight bound [108]. An alternative bound for individual measurements was introduced by Gill and Massar [130]. However, as this bound is, in general, less tight than the NHB, we shall not consider it in this work.

There is a hierarchy between the bounds described above, $C^{\mathrm{MI}}\geq N\geq H\geq \max\left(C^{\mathrm{RLD}},C^{\mathrm{SLD}}\right)$. However, it is known that for estimating a single parameter the SLD bound, the Holevo bound, the NHB and the most informative bound coincide, $C^{\mathrm{MI}}=N=H=C^{\mathrm{SLD}}$ [66,131]. This gives a simple method of explicitly computing the achievable precision for estimating one parameter. Additionally, when estimating any number of parameters using pure states, the Holevo bound and the NHB are equal, N=H, i.e., the CC and CQ strategies are equivalent, as are the $C_{a}$$C_{a}$ and $C_{a}$$Q_{a}$ strategies. Finally, note that although the Holevo bound is tighter than the SLD bound, it has recently been proven that the difference between the two is at most a factor of 2 [70,71].

### 2.3. Quantum Channels

In this work, we consider estimating qubit rotations in a noisy channel. There are three rotation parameters we wish to estimate: $\theta_{x}$, $\theta_{y}$, and $\theta_{z}$. These three parameters describe the rotation amplitudes of the three axes of the Bloch sphere:(15)$$\begin{array}{c} U\left(\theta_{x},\theta_{y},\theta_{z}\right)=\exp\left(i\frac{\theta_{z}}{2}\sigma_{z}\right)\exp\left(i\frac{\theta_{y}}{2}\sigma_{y}\right)\exp\left(i\frac{\theta_{x}}{2}\sigma_{x}\right)\;, \end{array}$$
where $\sigma_{x}$, $\sigma_{y}$ and $\sigma_{z}$ are the Pauli matrices. We insist that the rotation amplitudes are small so that the order of the rotations does not matter. Note that this assumption of small rotations is relevant provided it is possible to perform some prior characterization of the quantity to be estimated, such as in Ref. [63]. We are interested in finding the optimal single-qubit and two-qubit probes for estimating these parameters. We shall consider the three cases as follows: (i) a single rotation when $\theta_{y}=\theta_{z}=0$, (ii) two rotations when $\theta_{z}=0$ and (iii) all three rotations present. Depending on the number of parameters, the optimal probe and measurement strategy will be different. For certain channels, when estimating a single parameter $\theta_{x}$, as we shall show later, any pure state in the y−z plane will be optimal.

The rotations transform an input quantum state ρ to the state $\rho_{\theta }=U\left(\theta_{x},\theta_{y},\theta_{z}\right)$$\rho U{\left(\theta_{x},\theta_{y},\theta_{z}\right)}^{†}$. In our model, this rotated state is then subject to some noise. A noisy quantum channel can be described in the operator-sum representation as a linear map acting on the density matrix of the state subjected to the channel,
(16)$$E_{\theta }\left(\rho \right)=\sum_{a=1}^{K}M_{a}\rho_{\theta }M_{a}^{†}\;,$$
where the operators, $M_{a}$, obey the completeness relation, $\sum_{a=1}^{K}M_{a}^{†}M_{a}=𝟙$. In what follows, we shall drop the explicit dependence of the quantum channel on θ. The operators are known as operational elements of the channel, or Kraus operators [132]. Note that, as is evident from Equation (Equation 16), the overall quantum channel we consider involves rotating the quantum state before subjecting it to the noise. *K* is known as the Kraus number and satisfies $K\leq d^{2}$, where *d* is the dimensions of the system in question. For the map E(ρ) to represent a deterministic, physical channel, it must be linear, trace-preserving, and completely positive [133]. A deterministic channel here means that we do not allow postselection on certain measurement results. Quantum channels such as this are also called ‘trace-preserving completely positive’ maps. There are two distinct scenarios when considering estimation in the presence of noise. One is to perform the estimation using noisy probe states, E(ρ), i.e., the decoherence happens before the rotations. The other is to use pure probe states, where the decoherence occurs after the rotation. In this work, we consider the second option. In all cases, we assume that the noise parameter is known, i.e., we do not need to treat it as a nuisance parameter as has been considered before [134,135].

## 3. Results

We shall now present our results on the optimal estimation variances for three channels: (1) decoherence channel, (2) amplitude damping channel, and (3) phase damping channel. Figure 2 shows the effect each channel has on the Bloch sphere. These three channels have been considered before in the context of the QFI [136,137], and their physical motivation includes energy dissipation and decoherence in trapped ions [138,139,140,141]. For each channel, we present the optimal achievable precision for estimating one and two-qubit rotations with single-qubit probes and one-, two- and three-qubit rotations with two-qubit probes. For single-parameter estimation with single-qubit probes, the optimal probe state is determined from the SLD bound, as in Equation (Equation 7). In all other scenarios, we find the optimal probe numerically. We compare the performance of using an entangled two-qubit probe, where only one qubit passes through the channel, with that of having only a single-qubit probe. Thus, in our work, computing the NHB for the single-qubit and two-qubit probe states corresponds to the CC and $C_{a}$$C_{a}$ strategies, respectively. Computing the Holevo bound for the single-qubit and two-qubit probe states corresponds to the CQ and $C_{a}$$Q_{a}$ strategies, respectively. When we wish to compute the precision attainable with collective measurements on a finite number of copies, *M*, of the probe state, we shall evaluate the NHB for the state $\rho^{⊗M}$, scaled by a factor of *M* to ensure a fair comparison in terms of resources used. Note that the Holevo bound for the state $\rho^{⊗M}$, scaled by a factor of *M*, also provides a bound on the precision attainable with collective measurements on $\rho^{⊗M}$. However, this bound is only guaranteed to be tight as M→∞. In many of the problems we consider we are able to obtain analytic solutions for the matrices which optimize the Holevo and Nagaoka bounds based on the symmetries of the system. However, an analytic solution was not always possible and in this case, a combination of numerics and guesswork was used to obtain results.

### 3.1. Decoherence Channel

We first explicitly compute the achievable precision for individual and collective measurements with one and two-qubit probes under the action of a decoherence channel. This channel is represented by the following Kraus operators
(17)$$\begin{array}{c} \begin{array}{ccccccc} M_{0} & = & \sqrt{1-\frac{3\epsilon }{4}}𝟙 & , & M_{1} & = & \sqrt{\frac{\epsilon }{4}}\sigma_{x}\;,\\ M_{2} & = & \sqrt{\frac{\epsilon }{4}}\sigma_{y} & , & M_{3} & = & \sqrt{\frac{\epsilon }{4}}\sigma_{z}\;, \end{array} \end{array}$$
where 0≤ϵ≤1 parametrises the decoherence strength. These Kraus operators act only on the probe qubit as we wish to consider the situation where only a single qubit passes through the channel, as shown in Figure 1.

#### 3.1.1. Single-Qubit Probe

With a single-qubit, the optimal probe for sensing a rotation about the *x*-axis is any pure state in the Y-Z plane of the Bloch sphere. This is easily verified since, for estimating a single parameter, the Holevo bound coincides with the SLD bound, as in Equation (Equation 7). For the probe $|\psi \rangle =\cos\left(\theta /2\right)|0\rangle +e^{i\phi }\sin\left(\theta /2\right)|1\rangle$ and a particular choice of the decoherence parameter, ϵ=0.05, we show the Holevo bound as a function of the Bloch sphere angles, θ and ϕ, in Figure 3i. This figure verifies our claim that any probe in the Y-Z plane (ϕ=π/2,3π/2) is optimal. The computation of the SLD bound is described in Appendix A. We now consider the state |0〉, one of the many possible optimal states. After the decoherence channel, this probe is left in the state
(18)$$\rho =|0\rangle \langle 0|\left(1-\epsilon \right)+\frac{\epsilon }{2}\;.$$

For a single-qubit the optimal $X_{x}$ matrix for estimating a rotation about the *x*-axis is
(19)$$X_{x}=\frac{i}{1-\epsilon }\left(|0\rangle \langle 1|-|1\rangle \langle 0|\right)\;.$$

We show how $X_{x}$ is computed in Appendix B. Using this matrix, the Holevo bound is given by
(20)$$v_{x}\geq H_{1q,1}^{d}=\frac{1}{{\left(1-\epsilon \right)}^{2}}\;.$$

We use the superscript ‘d’ to denote the decoherence channel, the first subscript ‘1q’ to denote a one-qubit probe, and the second subscript ‘1’ to denote single parameter estimation. For single-parameter estimation, an individual measurement provides as much information as a collective measurement [122]. Hence
(21)$$\begin{array}{c} v_{x}\geq N_{1q,1}^{d}=\frac{1}{{\left(1-\epsilon \right)}^{2}}\;, \end{array}$$
where we use N to indicate the Nagaoka bound. In Appendix C we show an explicit measurement reaching this bound. We now consider estimating two parameters, rotations about the *x* and *y* axes. The |0〉 probe is still optimal as shown in Figure 3ii. The optimal precision with individual measurements can be obtained using the Nagaoka bound and is given by
(22)$$v_{x}+v_{y}\geq N_{1q,2}^{d}=\frac{4}{{\left(1-\epsilon \right)}^{2}}\;,$$
which is exactly four times the single parameter Nagaoka bound. This implies that an optimal strategy for estimating both parameters is to estimate each parameter separately: use N/2 probes to estimate $\theta_{x}$ and the remaining N/2 probes to estimate $\theta_{y}$. We show a measurement strategy reaching this bound in Appendix C. The *X*-matrix which is required to achieve this Nagaoka bound is given by
(23)$$X_{y}=\frac{1}{1-\epsilon }\left(|0\rangle \langle 1|+|1\rangle \langle 0|\right)\;.$$

The same *X*-matrix also optimizes the Holevo bound, giving
(24)$$v_{x}+v_{y}\geq H_{1q,2}^{d}=\frac{4-2\epsilon }{{\left(1-\epsilon \right)}^{2}}\;,$$
which is slightly smaller than the corresponding Nagaoka bound when ϵ is not equal to 0 or 1. Thus, in this case, we can achieve better precision by performing a collective measurement. The optimal strategy gives
(25)$$\begin{array}{c} v_{x}^{*}=v_{y}^{*}=\frac{1}{2}H_{1q,2}^{d}=\frac{2-\epsilon }{{\left(1-\epsilon \right)}^{2}}\;, \end{array}$$
which is always greater than $H_{1q,1}^{d}$. This indicates that by estimating the second parameter we lose some precision in our estimate of the first parameter as can be expected.

#### 3.1.2. Two-Qubit Probe

For estimating one, two, or three rotations with a two-qubit probe, we numerically verify that any maximally entangled two-qubit probe is equally optimal. Note that, in a slightly different setting, the maximally entangled state was proven to be optimal in the noiseless case [142]. We consider the following probe
(26)$$\begin{array}{cc} |\psi_{0}\rangle & =\frac{1}{\sqrt{2}}\left(|01\rangle +|10\rangle \right)\;. \end{array}$$

After passing through the channel the probe becomes
(27)$$\rho =\left(1-\epsilon \right)|\psi_{0}\rangle \langle \psi_{0}|+\frac{\epsilon }{4}𝟙\;.$$

For estimating one parameter with this probe the optimal *X* matrix is
(28)$$X_{x}=\frac{i}{\left(1-\epsilon \right)}\left(|\psi_{0}\rangle \langle \psi_{2}\left|-\right|\psi_{2}\rangle \langle \psi_{0}|\right)\;,$$
where
(29)$$\begin{array}{cc} |\psi_{2}\rangle & =\frac{1}{\sqrt{2}}\left(|00\rangle +|11\rangle \right)\;. \end{array}$$

The bounds for both variances, using individual and collective measurements, are given by
(30)$$v_{x}\geq H_{2q,1}^{d}=N_{2q,1}^{d}=\frac{2-\epsilon }{2{\left(1-\epsilon \right)}^{2}}\;,$$
which coincides with the SLD bound. In Appendix D we show that by performing a measurement on both qubits we can achieve this precision. We note that $H_{2q,1}^{d}\leq H_{1q,1}^{d}$ which indicates that using a two-qubit probe gives a better estimation precision even though the second probe does not go through the channel. We quantify the estimation precision as
(31)$$\begin{array}{c} \mathrm{precision}=\frac{1}{v_{x}^{*}}\;, \end{array}$$
where $v_{x}^{*}$ is the variance obtained from the optimal strategy. We compare single-qubit and two-qubit probes for estimating a single parameter in Figure 4.

We now proceed to estimating two parameters using an entangled two-qubit probe. Similar to the single parameter case, we can write
(32)$$\begin{array}{c} X_{y}=\frac{-1}{\left(1-\epsilon \right)}\left(|\psi_{0}\rangle \langle \psi_{3}\left|+\right|\psi_{3}\rangle \langle \psi_{0}|\right)\;, \end{array}$$
(33)$$\begin{array}{c} X_{z}=\frac{-i}{\left(1-\epsilon \right)}\left(|\psi_{1}\rangle \langle \psi_{0}\left|-\right|\psi_{0}\rangle \langle \psi_{1}|\right)\;, \end{array}$$
where
(34)$$\begin{array}{cc} |\psi_{1}\rangle & =\frac{1}{\sqrt{2}}\left(|01\rangle -|10\rangle \right)\;, \end{array}$$
(35)$$\begin{array}{cc} |\psi_{3}\rangle & =\frac{1}{\sqrt{2}}\left(|00\rangle -|11\rangle \right)\;, \end{array}$$
to arrive at
(36)$$\begin{array}{c} v_{x}+v_{y}\geq H_{2q,2}^{d}=\frac{2-\epsilon }{{\left(1-\epsilon \right)}^{2}}\;. \end{array}$$

The optimal variance of each parameter is
(37)$$\begin{array}{c} v_{x}^{*}=v_{y}^{*}=\frac{1}{2}H_{2q,2}^{d}=\frac{2-\epsilon }{2{\left(1-\epsilon \right)}^{2}}\;, \end{array}$$
which is exactly equal to the single-parameter result $H_{2q,1}^{d}$ in Equation (Equation 30). Hence, we find that with a two-qubit probe, we can estimate a second parameter without any degradation in the precision of the first. For the two-qubit probe, we are also able to compute the Nagaoka bound,
(38)$$v_{x}+v_{y}\geq N_{2q,2}^{d}=\frac{4-\epsilon }{2{\left(1-\epsilon \right)}^{2}}\;.$$

For a general ϵ, this Nagaoka bound is larger than the corresponding Holevo bound, indicating that an individual measurement is inferior to a collective measurement. In Appendix D, we construct a measurement scheme that saturates this bound.

Finally, for estimating three parameters, with the optimal $X_{z}$ matrix given in Equation (Equation 33), we find that
(39)$$\begin{array}{c} v_{x}+v_{y}+v_{z}\geq H_{2q,3}^{d}=\frac{6-3\epsilon }{2{\left(1-\epsilon \right)}^{2}}\;, \end{array}$$
and the optimal variance of each parameter is
(40)$$\begin{array}{c} v_{x}^{*}=v_{y}^{*}=v_{z}^{*}=\frac{1}{3}H_{2q,3}^{d}=\frac{2-\epsilon }{2{\left(1-\epsilon \right)}^{2}}\;. \end{array}$$

This is once again equal to the single-parameter result $H_{2q,1}^{d}$ in Equation (Equation 30). Hence, we find that with a two-qubit probe, we can estimate all three parameters simultaneously just as well as we can estimate just one parameter. The ability of entangled probe states to avoid trade-offs in multiparameter estimation was observed before [35,55,56]. However, when restricted to individual measurements, the NHB is given by
(41)$$\begin{array}{c} v_{x}+v_{y}+v_{z}\geq N_{2q,3}^{d}=\frac{3}{{\left(1-\epsilon \right)}^{2}}\;, \end{array}$$
and the optimal variance of each parameter is
(42)$$\begin{array}{c} v_{x}^{*}=v_{y}^{*}=v_{z}^{*}=\frac{1}{3}N_{2q,3}^{d}=\frac{1}{{\left(1-\epsilon \right)}^{2}}\;. \end{array}$$

We see that when we are restricted to individual measurements the estimation of another parameter further degrades measurement precision. The same *X* matrices which optimise the Holevo bound, optimise the NHB. In Appendix E, we show that there exists a measurement strategy that saturates the NHB in this case. The differences between individual and collective measurement precision are highlighted in Figure 5. This figure shows the hierarchy between the different schemes mentioned earlier in the text: $C_{a}$$Q_{a}$ $\geq \{C_{a}C_{a},\mathrm{CQ}\}\geq$ CC. For this particular example we find that $C_{a}$$C_{a}$ ≥ CQ.

#### 3.1.3. Decoherence of Both Qubits

Thus far, we have considered the case where the second qubit experiences no decoherence. This lack of decoherence is equivalent to storing the second qubit in a perfect quantum memory, something which is not feasible with current technology. Thus, we now consider the channel where both qubits in the two-qubit probe experience some decoherence. We expose the two-qubit probe to the channel where the first and second qubit experience decoherence amplitudes of $\epsilon_{1}$ and $\epsilon_{2}$, respectively. Under the action of this channel, for estimating either one or two parameters, the maximally entangled two-qubit probe achieves a Holevo bound of
(43)$$H_{2q,1}^{2d}=\frac{1}{2}H_{2q,2}^{2d}=\frac{1-\frac{1}{2}\left(\epsilon_{1}+\epsilon_{2}\right)+\frac{1}{2}\epsilon_{1}\epsilon_{2}}{{\left(1-\epsilon_{1}\right)}^{2}{\left(1-\epsilon_{2}\right)}^{2}}\;.$$

Although we have shown the advantages offered by two-qubit probes over single-qubit probes, this expression highlights the dangers associated with using highly entangled probes for estimation. Equation (Equation 43) is symmetric in $\epsilon_{1}$ and $\epsilon_{2}$, which is somewhat surprising given that the rotation we are trying to estimate acts on the first qubit only. In spite of this, the decoherence of the second qubit is equally damaging to our estimation ability. We see that when the second qubit is fully decohered we are unable to estimate with any precision at all, regardless of the decoherence of the first qubit. This is shown in Figure 6.

Perhaps the most physically relevant scenario is the one in which both qubits experience the same decoherence, i.e., $\epsilon_{1}=\epsilon_{2}$. In this situation for estimating a single parameter the single-qubit probe described in Section 3.1.1 always outperforms the two-qubit probe considered. For estimating two parameters with the two-qubit probe the Holevo bound, Equation (Equation 43), remains unchanged. Even though, when using the maximally entangled two-qubit probe, we can estimate a second parameter for free, when the noise in the system is sufficiently high the single-qubit probe can still outperform the two-qubit probe for estimating two parameters. Thus, we can see that the optimal probe to use in this instance depends on the decoherence amplitudes experienced by both qubits. It is worth noting that in this high noise regime, a different two-qubit probe will be optimal and the optimized two-qubit probe will always perform better than or equal to the optimized single-qubit probe. The fact that the maximally entangled two-qubit probe is no longer optimal is a reflection of the fact that highly entangled states are very susceptible to loss and noise, see Refs. [6,143].

#### 3.1.4. Collective Measurements on Multiple Copies of the State

Before concluding discussions on the decoherence channel, we consider what happens when we can perform collective measurements on *M* copies of the single-qubit probe, i.e., we have the state $\rho^{⊗M}$ available to measure. As M→∞, we expect to find $N_{1q^{⊗M},2}^{d}\to H_{1q^{⊗M},2}^{d}=\frac{1}{M}H_{1q,2}^{d}$, where we have used the additivity of the Holevo bound [144]. Let us first consider M=2. With two copies the probe becomes $\rho^{⊗2}=\mathrm{diag}\left\{{\left(1-\frac{\epsilon }{2}\right)}^{2},\frac{\epsilon }{2}\left(1-\frac{\epsilon }{2}\right),\frac{\epsilon }{2}\left(1-\frac{\epsilon }{2}\right),\frac{\epsilon^{2}}{4}\right\}$. The derivatives of this matrix with respect to the parameters of interest are given by:(44)$$\begin{array}{c} \frac{\partial \rho_{\theta }}{\partial \theta_{x}}=\left(\begin{array}{cccc} 0 & iA & iA & 0\\ -iA & 0 & 0 & iB\\ -iA & 0 & 0 & iB\\ 0 & -iB & -iB & 0 \end{array}\right)\;\mathrm{and}\;\frac{\partial \rho_{\theta }}{\partial \theta_{y}}=\frac{1}{4}\left(\begin{array}{cccc} 0 & A & A & 0\\ A & 0 & 0 & B\\ A & 0 & 0 & B\\ 0 & B & B & 0 \end{array}\right)\;, \end{array}$$
where $A=\frac{1}{2}\left(1-\epsilon \right)\left(1-\frac{\epsilon }{2}\right)$ and $B=\frac{1}{4}\left(1-\epsilon \right)\epsilon$. For this state the optimal $X_{x}$ and $X_{y}$ matrices are given by
(45)$$\begin{array}{c} X_{x}=\frac{i}{2(1-\epsilon )}\left(\begin{array}{cccc} 0 & 1 & 1 & 0\\ -1 & 0 & 0 & 1\\ -1 & 0 & 0 & 1\\ 0 & -1 & -1 & 0 \end{array}\right)\;\mathrm{and}\;X_{y}=\frac{1}{2(1-\epsilon )}\left(\begin{array}{cccc} 0 & 1 & 1 & 0\\ 1 & 0 & 0 & 1\\ 1 & 0 & 0 & 1\\ 0 & 1 & 1 & 0 \end{array}\right)\;, \end{array}$$
which give a Nagaoka bound of $N_{1q^{⊗2},2}^{d}=\left(2-\epsilon +\frac{\epsilon^{2}}{2}\right)/{\left(1-\epsilon \right)}^{2}$. Therefore, it is clear that with just two copies of the probe, the Nagaoka bound is close to the Holevo bound. A similar result was observed recently for optical magnetometry systems [145] and a measurement saturating the two-copy Nagaoka bound has been implemented experimentally [115]. In Appendix F, we show a measurement scheme that attains this bound. With the development of recent techniques, it is now possible to compute the Nagaoka bound efficiently [106]. This allows us to compute the Nagaoka bound for many copies of the single-qubit probe. In Figure 7, we compare the Holevo bound to the Nagaoka bound for an increasing number of copies of the single-qubit probe. We plot the difference in achievable precision as a measure of how close the two bounds are. As expected, with an increasing number of copies of the probe state, the Nagaoka bound tends to the Holevo bound. However, these results are purely theoretical, and in an experimental implementation with current capabilities such precision can only be reached for measurements on a limited number of copies of the probe state [115]. Note that if we are restricted to performing separable measurements (non-entangling POVMs) on the state $\rho^{⊗M}$, it is known that there is no advantage compared to individual measurements [146].

### 3.2. Amplitude Damping Channel

We now consider an amplitude-damping channel. This channel models the decay of a two-level atom from the excited state to the ground state. The Kraus operators for this channel are:(46)$$M_{0}=\left(\begin{array}{cc} 1 & 0\\ 0 & \sqrt{1-p} \end{array}\right)\;,\;M_{1}=\left(\begin{array}{cc} 0 & \sqrt{p}\\ 0 & 0 \end{array}\right)\;.$$

These operators model an atom, which, if it is in the excited state, will decay to the ground state with probability *p*, and if it is in the ground state will remain unaffected. Thus, when we consider this channel, we obtain significantly different variances depending on the probe we chose for the single-qubit case. For example, the |0〉 and |1〉 states, corresponding to the ground and excited states, respectively, will behave differently depending on the decoherence amplitude. If we wish to consider the time evolution of an atomic system, we can imagine applying these operators to our quantum state once per time interval. In each time interval, the atomic system has a certain probability of decaying and as t→∞ all of the atoms end up in the ground state.

#### 3.2.1. Single-Qubit Probe

Similar to the decoherence channel in the single-qubit case, the optimal probe for sensing a rotation about the *x*-axis is any pure state which lies in the Y-Z plane of the Bloch sphere, shown in Figure 8i. These probes are able to estimate a single parameter with a Holevo bound of
(47)$$v_{x}\geq H_{1q,1}^{\mathrm{am}}=\frac{1}{1-p}\;.$$

At first sight, it might seem strange that the state |0〉 which is unaffected by the channel will perform just as well as |1〉 which decays through the channel. The reason for this is that the probe undergoes rotation before the decoherence. After the information has already been encoded onto the probe, the amplitude-damping channel destroys the information at the same rate for both probes. This can be seen from the derivatives
(48)$$\begin{array}{c} \frac{\partial }{\partial \theta_{x}}E\left(|0\rangle \langle 0|\right)=-\frac{\partial }{\partial \theta_{x}}E\left(|1\rangle \langle 1|\right)=-\frac{\sqrt{1-p}}{2}\sigma_{y}\;, \end{array}$$
which only differ in sign for the two probes. Here E represents the amplitude damping channel. The *X* matrix which saturates this Holevo bound is
(49)$$X_{x}=\pm \frac{1}{\sqrt{1-p}}\sigma_{y}\;,$$
where the + is for the |1〉 state and the − is for the |0〉 state. For estimating a single parameter using these probes the Holevo bound is equal to the SLD bound.

For two parameter estimation using individual measurements, the two probes |0〉 and |1〉 remain optimal, as shown in Figure 8ii. We find that for either probe, the Nagaoka bound is given by
(50)$$v_{x}+v_{y}\geq N_{1q,2}^{\mathrm{am}}=\frac{4}{1-p}\;,$$
which is exactly four times the optimal single-parameter estimate. In Appendix G, we describe the measurement strategy required to reach this bound. The optimal strategy is to use half of the probes to estimate $\theta_{x}$ and the remaining half to estimate $\theta_{y}$. The variance in each estimate is
(51)$$v_{x}^{*}=v_{y}^{*}=\frac{1}{2}N_{1q,2}^{\mathrm{am}}=\frac{2}{1-p}\;.$$

Interestingly, when allowing for collective measurements, the optimal probe for estimating a rotation about the *x* and *y* axes is the state |1〉, which now outperforms the state |0〉. This is very surprising since the |1〉 probe is affected by the channel, while the |0〉 probe is not. This can be viewed as *decoherence-assisted metrology*. We can understand this phenomenon by observing that although the state |0〉 does not experience any decoherence, the rotated state $|0\rangle +\left(\theta_{y}-i\theta_{x}\right)/2|1\rangle$ does experience decoherence. Figure 8iii depicts the difference between the two probes |0〉 and |1〉. The difference in the probe performance can be attributed to the difference in the partial derivatives of the probe after the channel. The Holevo bounds for these two probes apply in the asymptotic limit, but this limiting behavior is already present if we consider a collective measurement on two probes. We can see that
(52)$$\begin{array}{c} \frac{\partial }{\partial \theta_{x}}\left(\rho ⊗\rho \right)=\frac{\partial \rho }{\partial \theta_{x}}⊗\rho +\rho ⊗\frac{\partial \rho }{\partial \theta_{x}}\;, \end{array}$$
will be different for the probes |0〉 and |1〉 even if they have the same $\partial \rho /\partial \theta_{x}$. We will return to this in Section 4. The probe |1〉 achieves a Holevo bound for estimating two parameters of
(53)$$v_{x}+v_{y}\geq H_{1q,2}^{\mathrm{am}}=\{\begin{array}{ccc} 4 & \mathrm{for} & p\leq 1/2\;,\\ \frac{4p}{1-p} & \mathrm{for} & p>1/2\;. \end{array}\;.$$

(The Holevo bound obtained from the suboptimal probe |0〉 is 4/(1−p) which coincides with the result for an individual measurement. Thus, collective measurements do not provide any advantage when using the probe |0〉). For every 0<p<1, a collective measurement will give greater precision compared to an individual measurement. The minimum variance attained by the optimal probe |1〉 is given by
(54)$$v_{x}^{*}=v_{y}^{*}=\frac{1}{2}H_{1q,2}^{\mathrm{am}}=\{\begin{array}{ccc} 2 & \mathrm{for} & p\leq 1/2\;,\\ \frac{2p}{1-p} & \mathrm{for} & p>1/2\;. \end{array}\;.$$

This is plotted in Figure 9. The $X_{y}$ matrix required to saturate this bound is
(55)$$X_{y}=\frac{-1}{\sqrt{1-p}}\sigma_{x}\;.$$

#### 3.2.2. Two-Qubit Probe

We now consider the two-qubit case where the first qubit is exposed to the amplitude damping channel and the second qubit is passed through the identity channel. We consider the probe $\frac{1}{\sqrt{2}}(|0,1\rangle +\left|1,0\rangle \right)$ which exhibits some of the advantages offered by two-qubit probes. Using this probe, we are able to estimate a rotation around the *x*-axis with a Holevo bound given by
(56)$$v_{x}\geq H_{2q,1}^{\mathrm{am}}=N_{2q,1}^{\mathrm{am}}=\frac{1}{1-p}\;.$$

Again this coincides with the SLD bound. Unlike the decoherence channel, for estimating a single parameter under the action of this channel, the entangled two-qubit probe does not actually offer any advantage. For estimating two parameters using individual measurements, the Bell state is optimal and the minimum variance that can be obtained is
(57)$$v_{x}+v_{y}\geq N_{2q,2}^{\mathrm{am}}=\left\{\begin{array}{ccc} \frac{16\left(3-\sqrt{1-p}\right)\left(1-\sqrt{1-p}\right)}{p(8+p)(1-p)} & \mathrm{for} & 0<p\leq 2\sqrt{2}-2\;,\\ \frac{4p}{1-p} & \mathrm{for} & 2\sqrt{2}-2<p<1\;. \end{array}\right.$$

For all 0<p<1, this probe is always better than the optimal single-qubit probe. The *X* matrices required to saturate this bound when $p<2\sqrt{2}-2$ are
(58)$$\begin{array}{c} X_{x}=\left(\begin{array}{cccc} 0 & -ai & -bi & (-1+i)w\\ ai & 0 & 0 & ai\\ bi & 0 & 0 & bi\\ (-1-i)w & -ai & -bi & 0 \end{array}\right)\;\mathrm{and}\;X_{y}=\left(\begin{array}{cccc} 0 & -a & -b & (-1-i)w\\ -a & 0 & 0 & a\\ -b & 0 & 0 & b\\ (-1+i)w & a & b & 0 \end{array}\right)\;, \end{array}$$
where $a=2/(4\left(1-p\right)+\left(4-p\right)\sqrt{1-p})$, $b=2\left(3+p\right)/(\left(4+p\right)\left(1-p\right)+\left(4+2p\right)\sqrt{1-p})$ and $w=\sqrt{(a^{2}+b^{2})/2}$. When $p>2\sqrt{2}-2$ the required matrices are
(59)$$\begin{array}{c} X_{x}=\left(\begin{array}{cccc} 0 & -i & \frac{-i}{\sqrt{1-p}} & c_{14+}\\ i & c_{22} & -c_{23}i & \frac{ip}{2\sqrt{1-p}}-\sqrt{1-p}c_{34}\\ \frac{i}{\sqrt{1-p}} & c_{23}i & \frac{-c_{22}}{1-p} & \frac{ip}{2(1-p)}+c_{34}\\ c_{14-} & \frac{-ip}{2\sqrt{1-p}}-\sqrt{1-p}c_{34} & \frac{-ip}{2(1-p)}+c_{34} & -c_{44} \end{array}\right)\;, \end{array}$$
and
(60)$$\begin{array}{c} X_{y}=\left(\begin{array}{cccc} 0 & -1 & \frac{-1}{\sqrt{1-p}} & c_{14-}\\ -1 & -c_{22} & -ic_{23} & \frac{p}{2\sqrt{1-p}}-i\sqrt{1-p}c_{34}\\ \frac{-1}{\sqrt{1-p}} & ic_{23} & \frac{c_{22}}{1-p} & \frac{p}{2(1-p)}+ic_{34}\\ c_{14+} & \frac{p}{2\sqrt{1-p}}+i\sqrt{1-p}c_{34} & \frac{p}{2(1-p)}-ic_{34} & c_{44} \end{array}\right)\;, \end{array}$$
where $c_{14\pm }=\frac{-1\pm i}{\sqrt{2}}\sqrt{\frac{2-p}{1-p}}$, $c_{22}=\sqrt{\left(p^{2}+4p-4\right)/\left(2\left(2-p\right)\right)}$, $c_{23}=\sqrt{\left(p^{2}+4p-4\right)/\left(2\left(1-p\right)\left(2-p\right)\right)}$, $c_{34}=\sqrt{p^{2}+4p-4}/\left(2\left(1-p\right)\right)$ and $c_{44}=p\sqrt{p^{2}+4p-4}/\left(\sqrt{2(2-p)}\left(1-p\right)\right)$. These matrices are constructed such that $X_{x}$ and $X_{y}$ commute. A POVM can then be formed from the simultaneous eigenvectors of the two matrices which saturates the Nagaoka bound.

When allowing for collective measurements, the maximally entangled probe gives
(61)$$v_{x}+v_{y}\geq H_{2q,2}^{\mathrm{am}}\left(\mathrm{bell}\right)=\left\{\begin{array}{ccc} \frac{{\left(2-p\right)}^{2}}{2{\left(1-p\right)}^{2}} & \mathrm{for} & p\leq 2/3\;,\\ \frac{4p}{1-p} & \mathrm{for} & 2/3<p<1\;. \end{array}\right.$$

Plotting both $H_{2q,2}^{\mathrm{am}}$ and $H_{1q,2}^{\mathrm{am}}\left(\mathrm{bell}\right)$ in Figure 9, we see that at some values of *p*, the single-qubit probe |1〉 outperforms the maximally entangled probe. This means that the maximally entangled probe is not always the optimal two-qubit probe for sensing the channel. The optimal two-qubit probe depends on the noise in the channel. The optimal two-qubit probe can be written as |ψ〉=a|0,1〉+b|1,0〉, where *a* and *b* depend on *p*. With an optimized probe, we get
(62)$$v_{x}+v_{y}\geq H_{2q,2}^{\mathrm{am}}=\{\begin{array}{ccc} \frac{2}{1-p} & \mathrm{for} & p\leq 1/2\;,\\ \frac{4p}{1-p} & \mathrm{for} & 1/2<p<1\;. \end{array}$$

The maximally entangled probe is only optimal when p=0 and when p≥2/3. For p≥1/2 the probe |1,0〉, which performs identically to the single-qubit probe |1〉, is optimal. When p<1/2, the optimal two-qubit probe is
(63)$$\begin{array}{c} |\psi \rangle =\sqrt{\frac{1-2p}{2-2p}}|0,1\rangle +\frac{1}{\sqrt{2-2p}}|1,0\rangle \;. \end{array}$$

The optimal $X_{x}$ and $X_{y}$ matrices corresponding to this probe are
(64)$$\begin{array}{cc} X_{x} & =\frac{-i}{\sqrt{1-p}}\left(|0\rangle \langle 1|-|1\rangle \langle 0|\right)⊗\left|0\rangle \langle 0\right|-\frac{i\sqrt{1-2p}}{1-p}\left|1\rangle \langle 1\right|⊗\left(|1\rangle \langle 0|-|0\rangle \langle 1|\right)\;, \end{array}$$
(65)$$\begin{array}{cc} X_{y} & =-\frac{1}{\sqrt{1-p}}\left(|0\rangle \langle 1|+|1\rangle \langle 0|\right)⊗\left|0\rangle \langle 0\right|+\frac{\sqrt{1-2p}}{1-p}\left|1\rangle \langle 1\right|⊗\left(|1\rangle \langle 0|+|0\rangle \langle 1|\right)\;. \end{array}$$

By direct substitution, one can check that these matrices satisfy the unbiased conditions. By direct substitution, we obtain the *Z* matrix as
(66)$$\begin{array}{c} Z=\left(\begin{array}{cc} \frac{1}{1-p} & 0\\ 0 & \frac{1}{1-p} \end{array}\right)\;, \end{array}$$
which gives H=2/(1−p). The attainable precision for estimating two parameters with both single- and two-qubit probes is plotted in Figure 9. Notably, this example shows that in different regimes, it is possible to have either $C_{a}$$C_{a}$>CQ $\left(1/N_{2q,2}^{\mathrm{am}}>1/H_{1q,2}^{\mathrm{am}}\right)$ or CQ>$C_{a}$$C_{a}$$\left(1/H_{1q,2}^{\mathrm{am}}>1/N_{2q,2}^{\mathrm{am}}\right)$.

Using the maximally entangled state as our probe, we find that the Holevo bound for estimating three parameters is given by
(67)$$v_{x}+v_{y}+v_{z}\geq H_{2q,3}^{\mathrm{am}}=\{\begin{array}{ccc} \frac{\left(3-2p)(2-p\right)}{2{\left(1-p\right)}^{2}} & \mathrm{for} & p\leq 2/3\;,\\ \frac{\left(2+7p\right)}{2-2p} & \mathrm{for} & 2/3<p<1\;. \end{array}$$

The optimal $X_{x}$, $X_{y}$ and $X_{z}$ matrices for this probe are
(68)$$\begin{array}{c} X_{x}=-i\left(\begin{array}{cccc} 0 & c_{1} & c_{2} & 0\\ -c_{1} & 0 & 0 & 0\\ -c_{2} & 0 & 0 & -c_{3}\\ 0 & 0 & c_{3} & 0 \end{array}\right)\;,\;X_{y}=-\left(\begin{array}{cccc} 0 & c_{1} & c_{2} & 0\\ c_{1} & 0 & 0 & 0\\ c_{2} & 0 & 0 & -c_{3}\\ 0 & 0 & -c_{3} & 0 \end{array}\right)\;\mathrm{and}\;X_{z}=i\left(\begin{array}{cccc} 0 & 0 & 0 & 0\\ 0 & 0 & -c_{2} & 0\\ 0 & c_{2} & 0 & 0\\ 0 & 0 & 0 & 0 \end{array}\right)\;, \end{array}$$
where
(69)$$\begin{array}{cc} c_{1} & =\{\begin{array}{ccc} \frac{p}{2-2p} & \mathrm{for} & 0\leq p\leq 2/3\;,\\ 1 & \mathrm{for} & 2/3<p<1\;, \end{array}\;, \end{array}$$
(70)$$\begin{array}{cc} c_{2} & =\frac{1}{\sqrt{1-p}}\;, \end{array}$$
(71)$$\begin{array}{cc} c_{3} & =\frac{1}{1-p}-c_{1}\;. \end{array}$$

We note that the maximally entangled probe is not the optimal probe for estimating three parameters under the action of this channel. In Figure 10, however, it can be seen that the maximally entangled probe achieves a variance that is very close to the numerically optimized Holevo bound.

#### 3.2.3. Decoherence of Both Qubits

If we consider the more realistic channel, where each qubit is individually subject to the amplitude damping channel, then the estimation performance can only at best match the performance of the channel where the second qubit is not affected by the channel. For estimating $\theta_{z}$ using a $\frac{1}{\sqrt{2}}\left(|0,1\rangle +|1,0\rangle \right)$ probe with both qubits exposed to separate amplitude damping channels, with probabilities $p_{1}$ and $p_{2}$, respectively, the Holevo bound is given by
(72)$$v_{z}\geq H_{2q,1}^{2\mathrm{am}}=\frac{1}{2-2p_{1}}+\frac{1}{2-2p_{2}}\;.$$

We can compare this to the Holevo bound for estimating $\theta_{z}$ using a single-qubit probe, which is given by Equation (Equation 47), if we orient the probe such that it is optimal for sensing a rotation about the *z*-axis. We again see that, although we are trying to estimate a rotation on the first qubit only, the Holevo bound is symmetric in $p_{1}$ and $p_{2}$. If the second qubit is maximally exposed to the amplitude damping channel the first qubit can no longer be used for parameter estimation. When both qubits are exposed to the same decoherence amplitude, $p_{1}=p_{2}$, the Holevo bound for the two-qubit probe is equal to the Holevo bound for the single-qubit probe. Thus, although more entangled probes offer some advantages under certain conditions, they also have weaknesses that single-qubit probes do not have, shown in Figure 11.

#### 3.2.4. Collective Measurements on Multiple Copies of the State

As with the decoherence channel, we now consider performing collective measurements on *M* copies of the single-qubit state. The probes |0〉 and |1〉 perform equally well for simultaneous estimation of two rotations about the *x* and *y* axes when we are restricted to individual measurements. Both probes give a Nagaoka bound of $N_{1q,2}^{\mathrm{am}}\left(|0\rangle \right)=N_{1q,2}^{\mathrm{am}}\left(|1\rangle \right)=4/\left(1-p\right)$. The Nagaoka bound is known to be attainable for a two-dimensional state. An optimal measurement is to perform a measurement of $\sigma_{x}$ half of the time and a measurement of $\sigma_{y}$ the other half of the time, as shown in Appendix G. However, we know from the Holevo bound, that when allowing for collective measurements on infinitely many probes, the probe |1〉 actually outperforms |0〉. For the probe |0〉, collective measurements do not provide any advantage over individual measurements, but for the probe |1〉, collective measurements improve the precision. We show how this is possible by constructing an explicit estimator to approach the Holevo bound. We first consider doing a collective measurement on the probe |1,1〉.

After the amplitude damping channel, the probe |1〉 transforms to the mixed state p|0〉〈0|+(1−p)|1〉〈1|. Two copies of this will be in the state $\rho_{1}=p^{2}\left|0,0\rangle \langle 0,0\right|+p\left(1-p\right)|0,1\rangle \langle 0,1|+p\left(1-p\right)|1,0\rangle \langle 1,0|+{\left(1-p\right)}^{2}\left|1,1\rangle \langle 1,1\right|$. As this is a mixed state, collective measurements may offer an advantage. By performing the optimization over the matrices *X* and *Y*, we find that the Nagaoka bound for the state |0,0〉 and |1,1〉 is
(73)$$\begin{array}{cc} N_{1q^{⊗2},2}^{\mathrm{am}}\left(|0,0\rangle \right) & =\frac{2}{1-p}\;, \end{array}$$
(74)$$\begin{array}{cc} N_{1q^{⊗2},2}^{\mathrm{am}}\left(|1,1\rangle \right) & =\frac{2}{1-p}-2p\;. \end{array}$$

The result for |0,0〉 is not surprising. As |0,0〉 is unaffected by this channel, it remains as a pure state, and hence collective measurements cannot offer any benefit. Therefore, the minimum variance is exactly half of the individual measurement case, meaning that the optimal estimation is to measure each probe individually. But the result for |1,1〉 is smaller than half of the individual measurement case. This is now a four-dimensional state and it is not certain that the Nagaoka bound can still be attained (Nagaoka conjectured it is always attainable, but only proved for certain specific cases). We construct an explicit measurement that saturates the bound in Appendix H, showing that the bound is attainable.

The optimal *X* and *Y* matrices that give the Nagaoka bound for |1,1〉 probe are
(75)$$\begin{array}{cc} X_{x} & =\frac{1}{\sqrt{1-p}}\left(|y+,y+\rangle \langle y+,y+|-|y-,y-\rangle \langle y-,y-|\right)\;, \end{array}$$
(76)$$\begin{array}{cc} X_{y} & =-\frac{1}{\sqrt{1-p}}\left(|x+,x+\rangle \langle x+,x+|-|x-,x-\rangle \langle x-,x-|\right)\;, \end{array}$$
where $|y\pm \rangle =\left(|0\rangle \pm i|1\rangle \right)/\sqrt{2}$ and $|x\pm \rangle =\left(|0\rangle \pm |1\rangle \right)/\sqrt{2}$ are the eigenvectors of $\sigma_{y}$ and $\sigma_{x}$, respectively (For the probe |0,0〉, the optimal matrices have the same form except for a sign change). In Figure 12, we show how the Nagaoka bound tends towards the Holevo bound for the probe |1〉, with an increasing number of copies of the probe. This shows the same effect as in Figure 7, but in a slightly different representation.

### 3.3. Phase Damping Channel

Finally, we consider the phase-damping channel. This channel describes the loss of information without the loss of energy and is unique to quantum systems. It is represented by the Kraus operators
(77)$$M_{0}=\sqrt{1-\frac{\epsilon }{2}}𝟙\;,\;M_{1}=\sqrt{\frac{\epsilon }{2}}\sigma_{z}\;.$$

This channel can be thought of as applying $\sigma_{z}$ with probability $\frac{\epsilon }{2}$. We can see why this channel is uniquely quantum mechanical by considering repeated applications of the channel to an arbitrary density matrix. If the initial quantum state is
(78)$$\rho_{0}=\left(\begin{array}{cc} a & b\\ b^{*} & c \end{array}\right)\;,$$
then after the action of this channel, the state becomes
(79)$$\rho_{\epsilon }=\left(\begin{array}{cc} a & (1-\epsilon )b\\ \left(1-\epsilon \right)b^{*} & c \end{array}\right)\;.$$

Repeated application of this channel is equivalent to allowing ϵ→1. The density matrix loses its off-diagonal elements and ends up as a completely diagonal matrix, i.e., the quantum state has become a completely classical mixture of |0〉’s and |1〉’s. In this way, the quantum mechanical properties of the system are lost. Thus, this channel does not have a direct classical analogue.

#### 3.3.1. Single-Qubit Probe

Geometrically this channel represents a loss of information about the *x* and *y* components of the Bloch vector. When restricted to real probes, for estimating a rotation about the *x*-axis the probes |0〉 and |1〉 perform identically under the action of this channel. Both probes are able to estimate a rotation about either the *x* or the *y*-axis individually with a Holevo bound given by Equation (Equation 20). This coincides with the SLD bound for these probes as is expected. The $X_{x}$ matrix required to saturate the Holevo bound is $\sigma_{y}/\left(1-\epsilon \right)$. However, by allowing complex probes the Holevo bound can be improved. The reason for this is that the phase-damping channel affects the Bloch sphere in an asymmetric way. One optimal probe is $|\psi \rangle =\frac{1}{2}\left((1+i)|0\rangle +(1-i)|1\rangle \right)$ which can achieve a Holevo bound of 1, no matter what the decoherence amplitude. The $X_{x}$ matrix required to saturate this is $X_{x}=-\left|0\rangle \langle 0\right|+\left|1\rangle \langle 1\right|$. However, as is evident from Figure 13i, almost any probe in the Y-Z plane of the Bloch sphere is optimal. For probes of the form $|\psi \rangle =\cos\left(\theta /2\right)|0\rangle +e^{i\phi }\sin\left(\theta /2\right)$, any state with ϕ=π/2 or 3π/2 will be optimal, except for states where θ is exactly equal to 0 or π. Similarly, if we want to estimate a rotation about the *y*-axis only any state with ϕ=0 or π will be optimal, except when θ is exactly equal to 0 or π. The discontinuity in the Holevo bound exactly at these extreme points was observed before and corresponds to a point where the rank of the state changes [147,148,149,150,151,152]. The small rotation that we are trying to estimate changes the states |0〉 or |1〉, which are are rank 1 states and are not decohered to states which have rank 2 and are decohered.

For estimating a rotation about the *x* and *y* axes simultaneously, consider the probe state $|\psi \rangle =a|0\rangle +\sqrt{1-a^{2}}|1\rangle$, with a=1−δ, where δ is small. With this probe as δ→0, the Holevo bound is given by
(80)$$v_{x}+v_{y}\geq H_{1q,2}^{\mathrm{pd}}=\{\begin{array}{ccc} 4 & \mathrm{for} & \epsilon \leq 1-1/\sqrt{2}\;,\\ \frac{1}{\left(2-\epsilon \right){\left(1-\epsilon \right)}^{2}\epsilon } & \mathrm{for} & \epsilon >1-1/\sqrt{2}\;. \end{array}$$

It might be expected that one could always estimate $\theta_{x}$ and $\theta_{y}$ with a minimum variance of 4, given that it is possible to estimate both individually with a variance of 1, simply by using half the probes to estimate $\theta_{x}$ and the other half to estimate $\theta_{y}$. However, Figure 13 shows why this is not possible. The optimal probes for estimating each angle are different and so the multiparameter estimation is degraded. The $X_{y}$ matrix required to saturate this Holevo bound is given by
(81)$$X_{y}=\left\{\begin{array}{ccc} \left(\begin{array}{cc} 0 & \frac{-1}{1-\epsilon }\\ \frac{-1}{1-\epsilon } & \frac{2}{a\sqrt{1-a^{2}}} \end{array}\right) & \mathrm{for} & \epsilon \leq 1-1/\sqrt{2}\;,\\ \frac{1}{\epsilon (2-\epsilon )}\left(\begin{array}{cc} 0 & -1+\epsilon \\ -1+\epsilon & \frac{1}{a\sqrt{1-a^{2}}} \end{array}\right) & \mathrm{for} & \epsilon >1-1/\sqrt{2}\;. \end{array}\right.$$

We see when a=0 or a=1 these matrices contain infinities, but this is not reflected in the QFI which is optimized as a→1. In fact, the matrix $X_{y}$ only satisfies the unbiased conditions in Equations (Equation 8) and (Equation 9), when δ→0. Interestingly exactly at δ=0, so that a=1, the Holevo bound is only $\frac{4}{{\left(1-\epsilon \right)}^{2}}$. This discontinuity is again explained by the changing rank of the state at this point and is shown in Figure 13iii.

For estimating the same two parameters with individual measurements we find that the Nagaoka bound is given by
(82)$$v_{x}+v_{y}\geq N_{1q,2}^{\mathrm{pd}}=\frac{{\left(2-\epsilon \right)}^{2}}{{\left(1-\epsilon \right)}^{2}}\;.$$

Thus, collective measurements offer an advantage over individual measurements in this particular instance. In Appendix J we present a measurement saturating the Nagaoka bound, and show that it requires an unequal number of measurements of the two parameters. This is due to the fact that when estimating these two parameters individually we do not obtain the same minimum variance. To saturate the Nagaoka bound we require the following $X_{y}$ matrix
(83)$$X_{y}=\frac{1}{\sqrt{1-a^{2}}}\left(\left(a-\frac{1}{a}\right)\left|0\rangle \langle 0\right|+a|1\rangle \langle 1|\right)\;.$$

Again there is a discontinuity exactly at a=1; in this case, the Nagaoka bound is $\frac{4}{{\left(1-\epsilon \right)}^{2}}$.

#### 3.3.2. Two-Qubit Probe

We note that some of the results presented for two-qubit probes under the action of this channel have already been presented in Ref. [106]; however, we include them here for completeness. We consider the scenario where the first qubit is subject to the phase damping channel and the second auxiliary qubit experiences only the identity channel. The probes |0,0〉 and |1,1〉 perform identically to their single-qubit counterparts. Thus, these probes are still unable to estimate a rotation about all three axes. This is to be expected since the probes are separable, hence the idler qubit has no effect.

For estimating a rotation about the *x*-axis the probe $\frac{1}{\sqrt{2}}\left(|0,1\rangle +|1,0\rangle \right)$ is optimal. When we estimate a rotation about either the *x*-axis or the *y*-axis individually using this probe the Holevo bound is independent of the channel parameter ϵ. The Holevo bound remains unaffected when we estimate a rotation about both the *x* and *y* axes simultaneously.
(84)$$v_{x}+v_{y}\geq H_{2q,2}^{\mathrm{pd}}=2\;.$$

This is optimised when $v_{x}^{*}=v_{y}^{*}=1$. Thus, the entangled probe is able to estimate a second parameter at no additional cost. The $X_{x}$ and $X_{y}$ matrices which saturate the Holevo bound are given by
(85)$$\begin{array}{cc} X_{x} & =i\left(|\psi_{0}\rangle \langle \psi_{2}\left|-\right|\psi_{2}\rangle \langle \psi_{0}\left|+\right|\psi_{1}\rangle \langle \psi_{3}\left|-\right|\psi_{3}\rangle \langle \psi_{1}|\right)\;, \end{array}$$
(86)$$\begin{array}{cc} X_{y} & =-\left(|\psi_{2}\rangle \langle \psi_{1}\left|+\right|\psi_{1}\rangle \langle \psi_{2}\left|+\right|\psi_{0}\rangle \langle \psi_{3}\left|+\right|\psi_{3}\rangle \langle \psi_{0}|\right)\;. \end{array}$$

For estimating these two parameters simultaneously, the Bell state is still optimal and the Nagaoka bound is given by:(87)$$v_{x}+v_{y}\geq N_{2q,2}^{\mathrm{pd}}=\frac{4}{2-\epsilon }\;.$$

The *X* matrices required to achieve the minimum possible variance using individual measurements are given by
(88)$$\begin{array}{c} X_{x}=\frac{2i}{2-\epsilon }\left(|\psi_{0}\rangle \langle \psi_{2}\left|-\right|\psi_{2}\rangle \langle \psi_{0}|\right)\;, \end{array}$$
(89)$$\begin{array}{c} X_{y}=\frac{-2}{2-\epsilon }\left(|\psi_{0}\rangle \langle \psi_{3}\left|+\right|\psi_{3}\rangle \langle \psi_{0}|\right)\;. \end{array}$$

Again, owing to the symmetry in the channel, the total variance is minimized when $v_{x}^{*}=v_{y}^{*}=\frac{1}{2}N_{2q,2}^{\mathrm{pd}}$. Thus, for estimating these two parameters individual measurements are not as powerful as collective measurements. For estimating a rotation about all three axes simultaneously this probe is able to achieve a Holevo bound of
(90)$$v_{x}+v_{y}+v_{z}\geq H_{2q,3}^{\mathrm{pd}}=2+\frac{1}{{\left(1-\epsilon \right)}^{2}}\;.$$

The $X_{z}$ matrix required to saturate this Holevo bound is
(91)$$X_{z}=\frac{i}{\left(1-\epsilon \right)}\left(|\psi_{0}\rangle \langle \psi_{1}\left|-\right|\psi_{1}\rangle \langle \psi_{0}|\right)\;.$$

The same $X_{z}$ matrix optimizes the NHB for estimating all three rotations simultaneously and we have
(92)$$v_{x}+v_{y}+v_{z}\geq N_{2q,3}^{\mathrm{pd}}=\frac{4}{2-\epsilon }+\frac{1}{{\left(1-\epsilon \right)}^{2}}\;.$$

In Appendix K, we construct a measurement scheme that saturates this bound. This channel showcases many of the benefits of using quantum resources in parameter estimation as summarised in Figure 14.

#### 3.3.3. Decoherence of Both Qubits

As with previous channels, if we consider the channel where both qubits are exposed to the phase-damping channel individually, the performance can only worsen compared to the original channel. However, as mentioned above when using the probe $\frac{1}{\sqrt{2}}\left(|0,1\rangle +|1,0\rangle \right)$ to estimate $\theta_{x}$ or $\theta_{y}$ we can achieve a Holevo variance of 1 regardless of the noise in the channel. This is still true when we consider decohering both qubits. Nevertheless, the effects of exposing the second qubit to the phase-damping channel can be examined by considering a rotation about the *z*-axis. This is shown in Figure 15 where we plot the Holevo bound for the $\frac{1}{\sqrt{2}}\left(|0,1\rangle +|1,0\rangle \right)$ probe with both qubits exposed to separate phase damping channels. Using this probe to estimate a rotation about the *z*-axis the Holevo bound is given by
(93)$$v_{z}\geq H_{2q,1}^{2\mathrm{pd}}=\frac{1}{{\left(1-\epsilon_{1}\right)}^{2}{\left(1-\epsilon_{2}\right)}^{2}}\;.$$

When $\epsilon_{2}=0$, we recover the case where the second qubit is exposed to the identity channel, but for all other values of $\epsilon_{2}$ the performance of this probe worsens. For this particular example, when the auxiliary qubit experiences no decoherence the single-qubit and two-qubit probes perform equally well. Thus, when the secondary qubit experiences any decoherence the two-qubit probe performs worse than the single-qubit probe. In this situation, which is more realistic, using an entangled two-qubit probe can bring a disadvantage, as the additional ancillary qubit introduces extra noise into the system which lowers the achievable precision.

#### 3.3.4. Collective Measurements on Many Copies of the State

Finally, we consider what happens when we can perform measurements on more than one copy of the single-qubit probe. Owing to the singularity when the single-qubit probe, $|\psi \rangle =a|0\rangle +\sqrt{1-a^{2}}|1\rangle$, has a=1, it is difficult to find analytic results for many copies of this probe. However, it is easy to numerically verify that with an increasing number of copies of the state the Nagaoka bound tends to the Holevo bound as expected. This is plotted in Figure 16.

## 4. Conclusions

This case study has examined the performance of single- and two-qubit probes for multiparameter estimation under the action of several quantum noise channels. The two-qubit probes that are considered offer increased robustness to environmental noise compared to single-qubit probes. It is shown that entanglement is a necessary resource to reach the ultimate limits in quantum metrology, required in both the state preparation and state measurement stages. Having the option of employing entanglement at both of these stages always results in a better or equivalent measurement precision than using entanglement at only one of these stages. However, there is no general hierarchy between using entanglement in the state preparation stage only and using entanglement in the measurement stage only. Indeed, different scenarios have been presented where each of these settings outperforms the other. We also found that while certain channels allow two-qubit probes to outperform single-qubit probes, these situations are somewhat unrealistic. In particular, under realistic channels, for a two-qubit probe to obtain the maximum advantage over single-qubit probes, we require that the second qubit is stored in a perfect quantum memory. If the quantum memory we store the second qubit in adds too much noise, while the first qubit passes through the channel, then the single-qubit probe can outperform certain two-qubit probes. Thus, we have shown that although entanglement in principle improves parameter estimation in qubit states, it also offers potential drawbacks under certain conditions. For each channel, we considered the attainability of the Holevo bound with a finite number of copies of the probe state. It was found that a collective measurement on a limited number of copies of the probe state was sufficient to attain a precision close to the Holevo bound. This suggests that the ultimate limits in quantum metrology may be approximately attainable in the near future.

There are several possible ways to extend the results in this paper. We have focused on estimating small fixed rotations, but in a more realistic scenario, the quantum channel may be dynamic, meaning that the rotations we wish to sense are continually changing. In this situation, quantum resources can still offer an advantage [20]. However, the assumption made in this paper that the angles are well known, i.e., we are performing local estimation, may no longer hold. Our results could be extended to this case through Bayesian estimation [153,154]. Finally, it would be interesting to investigate whether these results can be applied to quantum key distribution, where it is important to estimate the parameters of noisy channels [155,156,157].

## Figures and Tables

**Figure 1 entropy-25-01122-f001:**
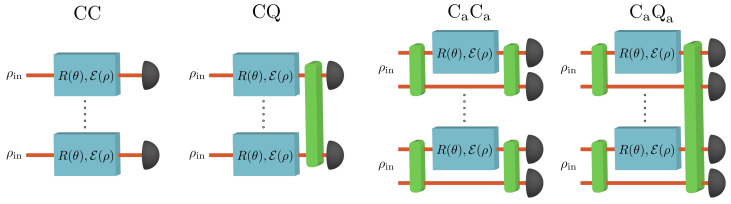
**The four sensing schemes we consider; Classical-Classical (CC), Classical-Quantum (CQ), Classical with ancilla-Classical with ancilla ($C_{a}$$C_{a}$) and Classical with ancilla-Quantum with ancilla ($C_{a}$$Q_{a}$).** The probes $\rho_{\mathrm{in}}$ are either unentangled single-qubit, or entangled two-qubit states, with the green box at the input symbolizing entanglement generation. After the probes have passed through the channel, we can either perform a collective measurement or an individual measurement, symbolized by the green box at the output. The $C_{a}$$C_{a}$ scheme has a green box at the output to signify that any measurement can be performed on the two-mode state, but collective measurements cannot be performed across multiple different states, as is the case with the CQ and $C_{a}$$Q_{a}$ schemes. The vertical dots in the schematic signify that many copies of the input probe state $\rho_{\mathrm{in}}$ are used. The blue box R(θ), E(ρ) represents the quantum channels. R(θ) is a small rotation (or multiple rotations) by an angle θ, the parameter we wish to measure, and E(ρ) is a noisy quantum channel.

**Figure 2 entropy-25-01122-f002:**
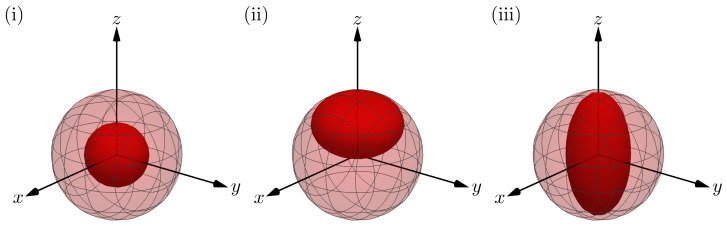
**Effect of the different quantum channels on the Bloch sphere**. The light red spheres and dark red spheroids represent the Bloch sphere for pure states before and after, respectively, passing through a decoherence channel parameterized by a decoherence strength of 0.5. The decoherence channel, amplitude damping channel, and phase damping channel are shown in Figures (**i**–**iii**), respectively.

**Figure 3 entropy-25-01122-f003:**
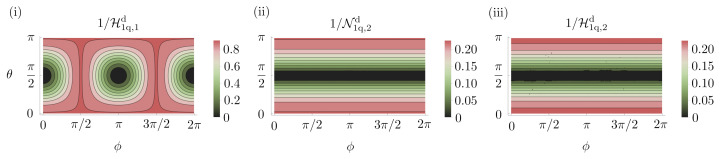
**Parameter estimation in a decoherence channel with single-qubit probes on the Bloch sphere, $|\psi \rangle =\cos\left(\theta /2\right)|0\rangle +e^{i\phi }\sin\left(\theta /2\right)|1\rangle$**. (**i**) Optimal achievable precision for estimating a single parameter obtained using the SLD bound, Equation (Equation 97). The optimal probes lie in the Y–Z plane of the Bloch sphere. (**ii**,**iii**) The optimal achievable precision for estimating two parameters simultaneously when using individual and collective measurements, respectively, was obtained using the semidefinite programs put forward in Refs. [106,118]. These plots show that the state |ψ〉=|0〉, which is one of many optimal states for estimating a single parameter, remains optimal for estimating two parameters. Contours are shown for ϵ=0.05.

**Figure 4 entropy-25-01122-f004:**
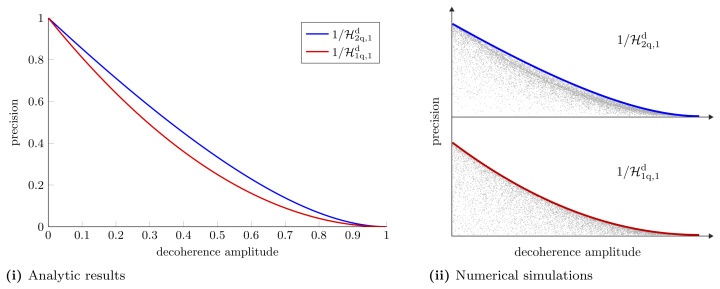
**Single parameter estimation in a decoherence channel.** (**i**) Optimal achievable precision for estimating a single parameter using one and two-qubit probes. These are given by the Holevo bounds Equations (Equation 20) and (Equation 30), respectively. (**ii**) Numerical simulations supporting the analytic results. For each plot in (**ii**) 10,000 random pure state probes are tested as the decoherence amplitude is varied from 0 to 1. The grey dots represent the precision from the random probes and the colored lines show the precision from the optimal probes.

**Figure 5 entropy-25-01122-f005:**
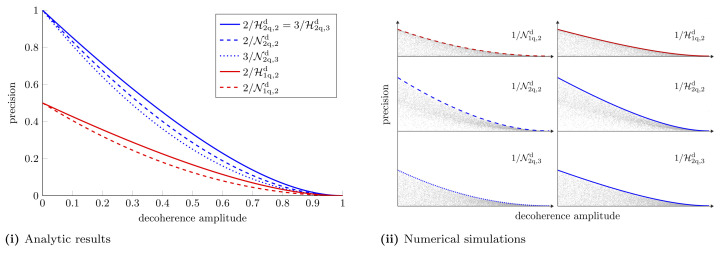
**Multiple parameter estimation in a decoherence channel.** (**i**) Holevo bounds and NHBs with one- and two-qubit probes for simultaneously estimating multiple parameters in a decoherence channel. With the single-qubit probe, for estimating two parameters, the Nagaoka bound and Holevo bound are given by Equations (Equation 22) and (Equation 24), respectively. For the two-qubit probe the Holevo and Nagaoka bounds for estimating two parameters are given by Equations (Equation 36) and (Equation 38), respectively. For estimating three parameters the Holevo bound and NHB are given by Equations (Equation 39) and (Equation 41), respectively. (**ii**) Numerical simulations to support analytic results. As before, each plot shows the precision attained with 10,000 random probes, which always lies below the precision attained with the optimal probe.

**Figure 6 entropy-25-01122-f006:**
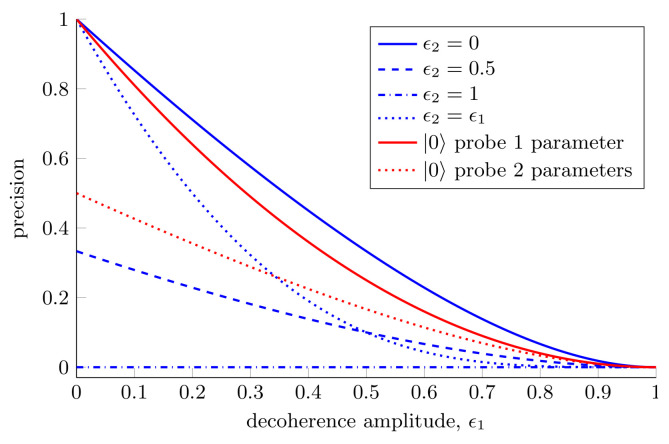
**Two parameter estimation in a decoherence channel affecting both qubits.** The precision that can be achieved with one- and two-qubit probes when simultaneously estimating one or two parameters in a channel where both qubits experience some decoherence. For the single-qubit probe the Holevo bounds for estimating one and two parameters are given by Equations (Equation 20) and (Equation 24), respectively. For the two-qubit probe, the Holevo bound is given by Equation (Equation 43).

**Figure 7 entropy-25-01122-f007:**
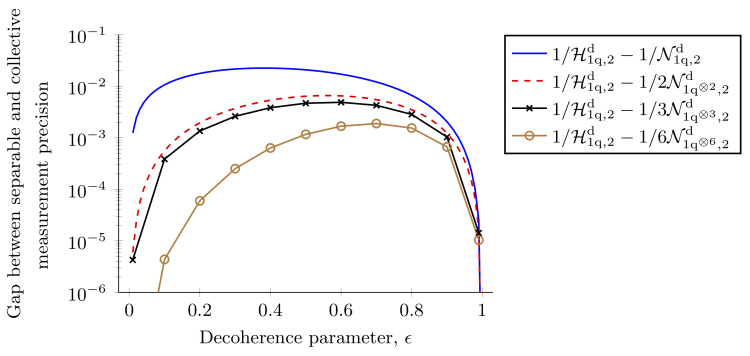
**Achievable precision using collective measurements versus individual measurements in the decohering channel.** Difference between the inverse of the Holevo bound for a single-qubit probe and the inverse of the Nagaoka bound for *M* copies of the same probe. As *M* increases the Nagaoka bound tends towards the Holevo bound as expected. For one and two copies of the probe, the Nagaoka bound is obtained analytically, for three and six copies of the probe we numerically obtain the Nagaoka bound for ϵ in the range 0.01→0.99.

**Figure 8 entropy-25-01122-f008:**
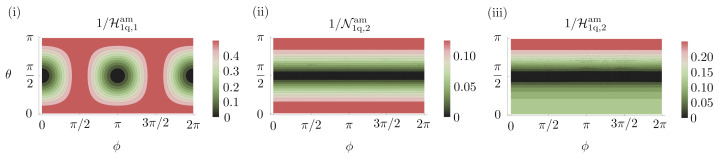
**Parameter estimation with a single-qubit probe in the amplitude damping channel with the probe $|\psi \rangle =\cos\left(\theta /2\right)|0\rangle +e^{i\phi }\sin\left(\theta /2\right)|1\rangle$.** (**i**) The precision achievable when estimating a rotation about the *x*-axis. (**ii**,**iii**) show the precision achievable for simultaneously estimating rotations about the *x* and *y*-axis when using individual measurements on each copy of the probe state and collective measurements on multiple copies of the probe state, respectively. For these figures p=0.5 and we can see that when estimating two parameters the |1〉 state (θ=π) outperforms the |0〉 state (θ=0) only if we allow for collective measurements. This is despite the fact that the |1〉 state experiences the decoherence whereas the |0〉 state does not.

**Figure 9 entropy-25-01122-f009:**
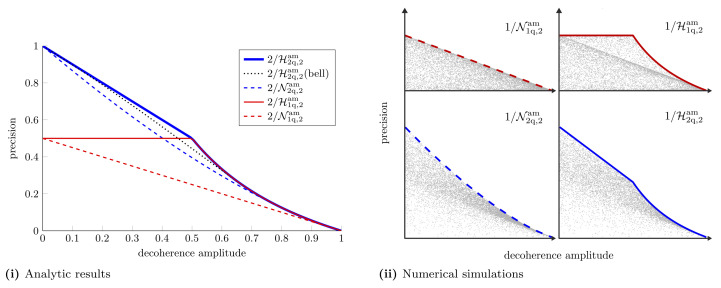
**Two parameter estimation in an amplitude damping channel.** (**i**) Achievable precision using single- and two-qubit probes to estimate two parameters under the amplitude damping channel. The Nagaoka bound and Holevo bounds for the single-qubit probe are given by Equations (Equation 50) and (Equation 53), respectively. They can be achieved when the probe is |1〉. For the two-qubit probe, the Nagaoka bound is given by Equation (Equation 57) and is achieved with a maximally entangled probe. The Holevo bound is given by Equation (Equation 62). The optimal probe depends on *p*. The Holevo bound for the maximally entangled probe Equation (Equation 61) is plotted in black for comparison. (**ii**) Numerical simulations which support the analytic results.

**Figure 10 entropy-25-01122-f010:**
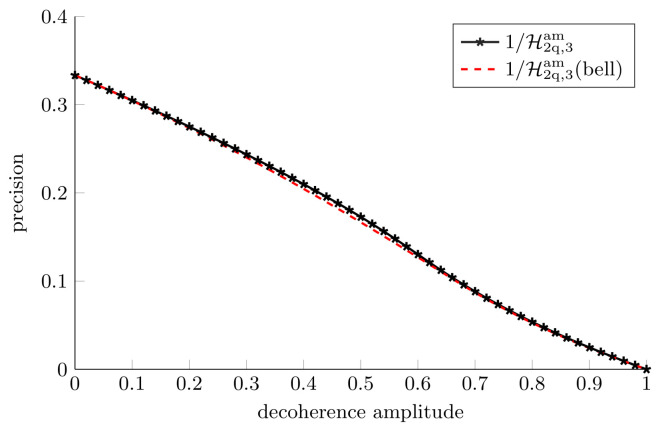
**Collective measurements for three parameter estimation in an amplitude damping channel.** Note that now the optimal $v_{x}=v_{y}\neq v_{z}$ and so instead we plot $1/H_{3}$. The optimal probe was found numerically. The Holevo bound for the Bell state, (Equation (Equation 67)) is almost optimal.

**Figure 11 entropy-25-01122-f011:**
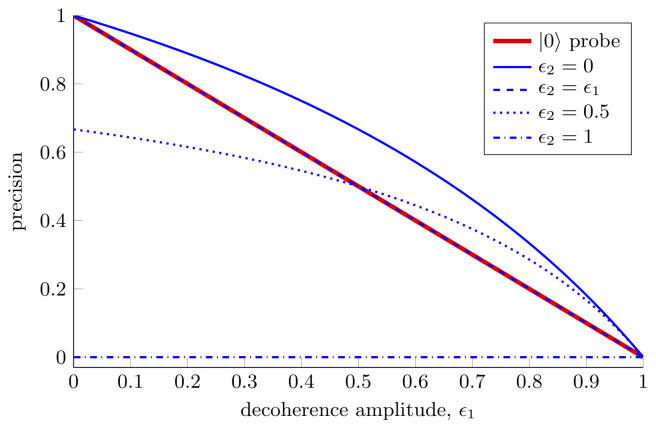
**Single parameter estimation in an amplitude damping channel where both qubits are exposed to the noise.** We plot the Holevo bounds achievable using single- and two-qubit probes to estimate a single parameter, $\theta_{z}$, with both qubits exposed to the amplitude damping channel. The Holevo bound for the single-qubit probe is given by Equation (Equation 47). For the two-qubit probe, the Holevo bound is given by Equation (Equation 72).

**Figure 12 entropy-25-01122-f012:**
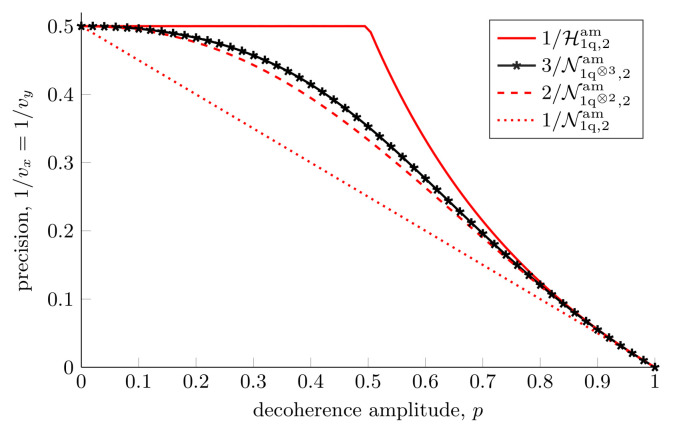
**Collective measurements for two parameter estimation in an amplitude damping channel.** Optimal achievable precision for estimating two parameters using the probe |1〉 when doing individual measurements $N_{1q,2}^{\mathrm{am}}$, Equation (Equation 57), and when measuring the probes in pairs, $N_{1q^{⊗2},2}^{\mathrm{am}}$, Equation (Equation 74). The Nagaoka bound for measuring three qubits collectively is plotted as $N_{1q^{⊗3},2}^{\mathrm{am}}$. We did not check that $N_{1q^{⊗3},2}^{\mathrm{am}}$ is attainable, but it is still a valid bound. Finally, Holevo’s bound applies when we are able to measure infinitely many copies simultaneously.

**Figure 13 entropy-25-01122-f013:**
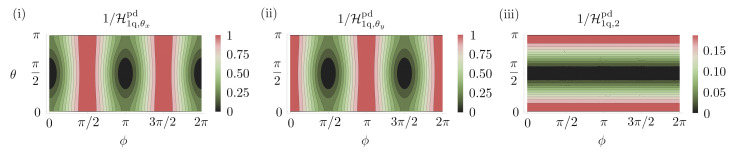
**Parameter estimation in the phase damping channel with the probe |ψ〉=cos(θ/2)|0〉$+e^{i\phi }\sin\left(\theta /2\right)|1\rangle$.** (**i**,**ii**) show the precision achievable when estimating a rotation about the *x*-axis and *y*-axis, respectively. We see that probes which are optimal for estimating $\theta_{x}$ are not optimal for estimating $\theta_{y}$. This explains why, although it is possible to estimate a single parameter without degradation in precision as the decoherence increases, the same is not possible when estimating two parameters. (**iii**) Holevo bound for estimating both rotations simultaneously with multiple copies of the same probe. There is a discontinuity in all figures at θ=0 or π. For all figures ϵ=0.5.

**Figure 14 entropy-25-01122-f014:**
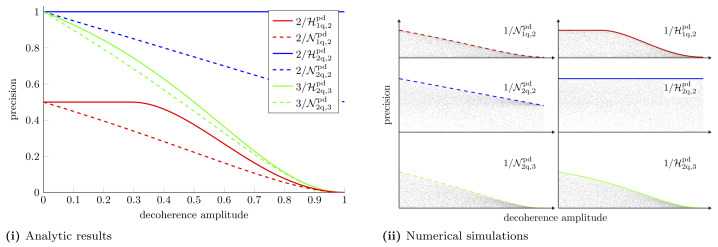
**Multiple parameter estimation in a phase damping channel.** (**i**) Achievable precision for various probes under the action of the phase damping channel. For estimating $\theta_{x}$ and $\theta_{y}$ simultaneously with the probe $\frac{1}{\sqrt{2}}[|0,1\rangle +\left|1,0\rangle \right]$, the Holevo bound is equal to 1 regardless of how noisy the channel is. The corresponding Nagaoka bound is given by Equation (Equation 87). For estimating a rotation about all three axes simultaneously, the Holevo and NHBs are given by Equations (Equation 90) and (Equation 92), respectively. For the |0〉 probe, estimating $\theta_{x}$ and $\theta_{y}$ simultaneously the Holevo bound is given by Equation (Equation 80) and the Nagaoka bound is given by Equation (Equation 82). (**ii**) Numerical simulations supporting the analytic results.

**Figure 15 entropy-25-01122-f015:**
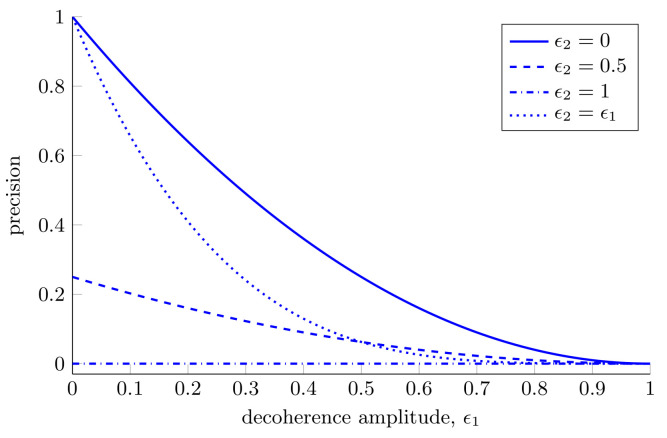
**Single parameter estimation in a phase damping channel.** Achievable precision for estimating $\theta_{z}$ using the $\frac{1}{\sqrt{2}}(|0,1\rangle +\left|1,0\rangle \right)$ probe with both qubits exposed to separate phase damping channels. The Holevo bound is given by Equation (Equation 93). When $\epsilon_{2}=0$ we obtain the achievable precision using the best possible single-qubit probe.

**Figure 16 entropy-25-01122-f016:**
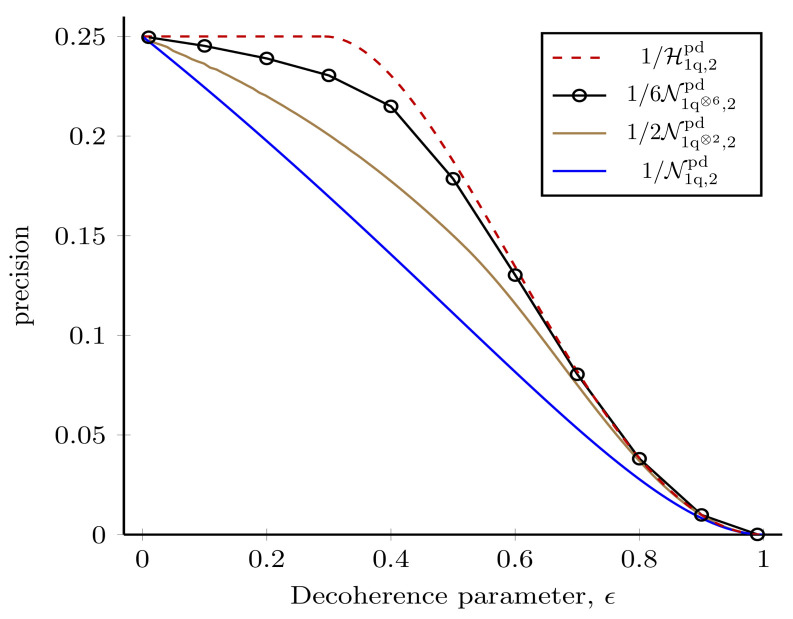
**Achievable precision using collective measurements versus individual measurements in the phase damping channel.** Achievable precision when performing a collective measurement on *M* copies of the same probe. As the number of copies of the probe increases, the Nagaoka bound tends toward the Holevo bound as expected. For a single copy of the probe, the Nagaoka bound is obtained analytically; for two and six copies of the probe, we numerically obtain the Nagaoka bound for ϵ in the range 0.01→0.99.

## Data Availability

All data are available upon reasonable request to the authors.

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
