# Peer review of "Multiparameter Estimation with Two-Qubit Probes in Noisy Channels"

_entropy, 2023, doi:10.3390/e25081122_

Round 1

Reviewer 1 Report

The work of Conlon and co-authors studies the performance of various qubit probe schemes in noisy channels and finds the optimal setup in the different scenarios. In the study, the noisy channels they considered are adequate and their mathematical deduction appears to be reliable. Importantly, they have shown many interesting and counterintuitive results such as the entanglement resource does not necessarily have an advantage in some scenarios, the trade-off between number and accuracy is not necessary for multiple-qubit probes, and decoherence is better. In general, I recommend the publication of the submitted work after the authors address the following concerns and comments.

1. In Fig.1, is it assumed that the rotation is constant for multiple rho_in? In practical systems, are the multiple rho_in sent one by one? Because the quantum channels are always dynamic, if more probs needs more time, is there a loss in precision? I notice the authors claim the rotation amplitudes are small, I suggest they clarify the practical scenarios that satisfy the assumption.

2. It is better to redraw Fig.2 because the figure looks 2D but the axes are arranged as 3D.

3. For different schemes of qubit probes, it is better to clarify whether the precision they obtained (e.g. Fig.6) is theoretical bound or experimental simulation. In practical operations, is there a difference between observable precision and calculated precision?

4. For counterintuitive results, the authors have provided sound mathematical deduction, but the explanation of the physical image is lacking.

5. I suggest the author discuss the relationship between qubit probes and quantum information processes. For example, characterizing the quantum channel is the task for quantum communication such as quantum key distribution. The interference visibility is sensitive to the rotation, noise and loss in channels (Nat. Photon. 16, 154–161 (2022), Photon. Res. 9, 1881-1891 (2021), Phys. Rev. Lett. 108, 130503 (2012). and Optica 4, 1016-1023 (2017)). Is it helpful to improve the performance of these quantum communication systems? It is better to discuss this in conclusion to extend the scenarios their results are effective in.

Reviewer 2 Report

The paper is well written and pedagogic. It can help the readers to understand some of the basic concepts about noisy channels.  I recommend the publication of the manuscript. 

Author Response

Thank you for taking the time to review our manuscript. We are glad that you think it is worthy of publication.

Round 2

Reviewer 1 Report

I appreciate the authors’ significant revision of the manuscript. All my concerns are addressed, therefore I recommend this manuscript for publication in Entropy.